# The Impact of Early Antiretroviral Treatment (ART) for HIV on the Sensitivity of the Latest Generation of Blood Screening and Point of Care Assays

**DOI:** 10.3390/v14071426

**Published:** 2022-06-29

**Authors:** Marion Vermeulen, Cari van Schalkwyk, Genevieve Jacobs, Karin van den Berg, Mars Stone, Sonia Bakkour, Brian Custer, Ute Jentsch, Michael P. Busch, Edward Murphy, Eduard Grebe

**Affiliations:** 1South African National Blood Service, Johannesburg 1715, South Africa; genevieve.jacobs@sanbs.org.za (G.J.); karin.vandenberg@sanbs.org.za (K.v.d.B.); ute.jentsch@sanbs.org.za (U.J.); 2School of Clinical Medicine, University of the Free State, Bloemfontein 9301, South Africa; 3National Research Foundation Centre of Excellence in Epidemiological Modelling and Analysis (SACEMA), The South African Department of Science and Innovation, Stellenbosch University, Stellenbosch 7602, South Africa; carivs@sun.ac.za (C.v.S.); egrebe@vitalant.org (E.G.); 4Division of Clinical Haematology, Deprtment of Medicine, University of Cape Town, Cape Town 7701, South Africa; 5Vitalant Research Institute, San Francisco, CA 94118, USA; mstone@vitalant.org (M.S.); sbakkour@vitalant.org (S.B.); bcuster@vitalant.org (B.C.); mbusch@vitalant.org (M.P.B.); emurphy@vitalant.org (E.M.); 6Department of Laboratory Medicine, University of California, San Francisco, CA 94143, USA

**Keywords:** antiretroviral therapy, HIV, point of care testing, serological screening

## Abstract

Introduction: Rapid initiation of antiretroviral therapy (ART) in early HIV infection is important to limit seeding of the viral reservoir. A number of studies have shown that if ART is commenced prior to seroconversion, the seroconversion may, or may not, occur. We aimed to assess whether seroreversion or no seroconversion occurs using samples collected during an early treatment study in South Africa. Methods: We tested 10 longitudinal samples collected over three years from 70 blood donors who initiated ART after detection of acute or early HIV infection during donation screening on fourth- and fifth-generation HIV antibody and RNA assays, and three point of care (POC) rapid tests. Donors were allocated to three treatment groups: (1) very early, (2) early, and (3) later. Longitudinal samples were grouped into time bins post-treatment initiation. Results: On all three high-throughput HIV antibody assays, no clear pattern of declining signal intensity was observed over time after ART initiation in any of the treatment initiation groups and 100% detection was obtained. The Abbott Determine POC assay showed 100% detection at all time points with no seroreversion. However, the Abbott ABON HIV1 and OraSure OraQuick POC assays showed lower proportions of detection in all time bins in the very early treated group, ranging from 50.0% (95% CI: 26.8–73.2%) to 83.1% (95% CI: 64.2–93.0%), and moderate detection rates in the early and later-treated groups. Conclusion: While our findings are generally reassuring for HIV detection when high-throughput serological screening assays are used, POC assays may have lower sensitivity for detection of HIV infection after early treatment. Findings are relevant for blood safety and other settings where POC assays are used.

## 1. Introduction

Rapid initiation of antiretroviral therapy (ART) in early HIV infection may limit the establishment of the HIV reservoir and may improve clinical outcomes. The acute phase of infection is associated with extremely high plasma viral loads which initially infect the gut mucosal CD4 cells causing inflammation, gut permeability, microbial translocation, and immune activation that further drives viral dissemination [1,2]. ART blocks reverse transcriptase to disrupt the replication of the HIV genetic code, and therefore, reduces the viral load to undetectable levels. Early ART initiation may preserve the gut mucosa and associated immune responses, and therefore, limit seeding of the viral reservoir [3]. For these reasons, a person diagnosed during acute infection should commence treatment urgently. Additionally, pre-exposure prophylaxis (PrEP) is recommended for at-risk groups.

Several studies have been initiated to identify early acute infections and start ART in the pursuit of a possible functional HIV cure [4]. These studies have shown that if ART is commenced during the primary infection stage, i.e., prior to seroconversion, that seroconversion may not occur, or conversely, seroreversion may occur in some cases when tested with both fourth generation HIV antibody/antigen assays, and with point-of-care (POC) rapid assays at around 24 weeks [5,6,7,8]. In addition, persons with known HIV infection may volunteer to donate blood while on ART in South Africa [9] and the United States [10], raising potentially serious concerns for blood safety. The nucleic acid test (NAT) may be negative due to viral suppression, and if treatment had been initiated early enough, individuals may never have seroconverted, or may have seroreverted. Such a donated unit of blood would, therefore, be made available for patient use, despite the donor being HIV infected.

At the South African National Blood Service (SANBS), blood is screened using high throughput serological and molecular assays in parallel, enabling the identification of HIV RNA and/or antibody (Ab) reactive donations [11]. Recently, the National Heart, Lung, and Blood Institute Recipient Epidemiology and Donor Evaluation Study (REDS-III) [12] conducted a prospective cohort study called the Monitoring and Acute Treatment of HIV Study (MATHS), in which a cohort of blood donors with acute (pre-seroconversion) and recent (determined by Limiting antigen (LAg) avidity) HIV was started immediately on ART, and followed clinically with longitudinal blood sampling.

We aimed to test the stored residual sample of each of these early treated samples using the latest fourth and fifth generation blood screening assays and three POC rapid tests used in South Africa for the diagnosis of HIV to assess whether seroreversion occurred and, if so, at what time after initiation of ART. We also analysed a set of cross-sectional HIV testing results from HIV positive donations found, retrospectively, to contain antiretroviral drug metabolites.

## 2. Methods

### 2.1. Setting

The South African National Blood Service collects approximately 900,000 blood donations per annum in a high HIV burden context, and screens individual donation samples in parallel for HIV RNA using NAT and anti-HIV Abs. Approximately 1600 confirmed HIV seropositive and 60 HIV NAT yield (NAT positive, Ab negative) donations are identified per year.

### 2.2. Routine Screening

During the study period, all blood donations were screened by the PRISM HIV O Plus (Abbott, Delkenheim, Germany) 3rd generation anti-HIV Ab assay, Hepatitis B surface antigen (Ag), and anti-HCV Ab chemiluminescent immunoassays (ChLIA), and, in parallel with the Procleix Ultrio (Plus) NAT assay (Grifols, Barcelona, Spain) for HIV RNA, HCV RNA, and HBV DNA. An extensive confirmatory and follow-up algorithm (previously described [11]) was in place for HIV identification, which included testing HIV NAT yields for p24 Ag and concordant (NAT and Ab) positive donations by Western blot to classify donations by Fiebig stage [13]. 

### 2.3. MATHS Cohort

For the MATHS study, we enrolled blood donors with acute infections who were identified as Fiebig stages I and II (NAT positive, antibody negative) into an early treatment study and followed them over a maximum three-year period. Blood samples were collected every two weeks during the first three months, then monthly until 6 months and then quarterly for the remaining 30 months. Later, we also enrolled blood donors who were identified as having recently acquired HIV infection by testing using the LAg avidity assay (Sedia Biosciences Corporation, Portland, OR, USA) and applying a normalized optical density (ODn) cut off of 1.5. Above an ODn of 1.5, the LAg assay detects antibodies that have matured and have high avidity through somatic hypermutation [14]. A total of 70 donors commenced treatment early during infection and provided 10 longitudinal samples each that were stored in the biorepository at SANBS. The median time to viral suppression for this cohort was 33 days (IQR 14–56 days). 

### 2.4. HIV Positive Donations Tested for ART Drugs

In a separate study [15], we retrospectively tested all HIV positive donations that were positive for HIV RNA and/or anti-HIV Ab in 2017, and, for the presence of anti-retroviral (ARV) drug metabolites using liquid chromatography-mass spectrometry (LC-MS). Furthermore, since the discovery of blood donors not disclosing their HIV status or ARV use [9,15], we operationalised testing in 2020 for the presence of ARV drug metabolites in all donations that tested anti-HIV positive only (serology yield). In this study, we evaluated the distributions of signal intensities observed using the primary serological screening assay—Abbott PRISM for the 2017 donations, and Abbott Alinity for the 2020 donations. 

### 2.5. Assays 

Each stored longitudinal sample from the MATHS study was tested for anti-HIV Abs on three high throughput blood-screening assays: the Abbott HIV Ag/Ab Combo on the Alinity s platform (Abbott Alinity) (Abbott, Wiesbaden, Germany), the Roche Elecsys HIV Duo on the cobas e801 platform (Roche HIV Duo) (Roche, Pleasanton, CA, USA), and the Bio-Rad BioPlex 2200 HIV Ag-Ab assay on the BioPlex 2200 system (Bio-Rad BioPlex) (Biorad, Hercules, CA, USA), and three POC assays: the Abbott Determine HIV-1/2 Ag/Ab Combo (Abbott Determine) (Abbott, Wiesbaden, Germany), the Abbott ABON HIV 1/2/O Tri-Line Human Immunodeficiency Virus Rapid Test (Abbott ABON HIV) (Abbott, Wiesbaden, Germany), and the OraSure OraQuick ADVANCE Rapid HIV-1/2 Antibody Test (OraSure OraQuick) (Orasure Technologies, Bethlehem, PA, USA).

The Abbot Alinity is a 4th-generation Ag/Ab combo assay that detects both antigen and antibody without the ability to discriminate between the two. The Roche HIV Duo and Bio-Rad Bioplex 2200 are pseudo fifth generation assays. The mechanism of detection is the same as the 4th-generation assays however the antigen and antibody reactions are separate allowing for discrimination.

The rapid POC assays are not licensed for blood screening. The Abbott Determine detects antigen and antibody whereas the Abbott ABON HIV and OraSure OraQuick only detect antibody. These POC assays are immunochromatographic lateral flow assays that use a test strip that is pre-coated with HIV antigens. 

### 2.6. Statistical Analysis

We stratified donors into three groups according to time from infection to treatment initiation, as well as Fiebig stage at treatment initiation. Estimated Dates of Detectable Infection (EDDI) were estimated using the routine screening test results, in the event the donor had not fully seroconverted at the time of the index donation using previously described methods [16]. If all routine screening tests were positive, donors were assigned an approximate duration of infection based on the observed LAg Avidity EIA normalised optical density (ODn) equal to the mean duration of recent infection associated with that ODn value when treated as a recency discrimination threshold. The groups were “Very early treatment” (≤30 days from EDDI to treatment initiation), “Early treatment” (31 to 60 days), and “Later treatment” (>60 days). 

Quantitative signal intensity (expressed as signal to cut-off ratio or S/CO) and qualitative results were analysed for high-throughput blood screening assays, and qualitative results for POC assays. Signal intensity over time since treatment initiation did not exhibit any clear pattern in reactivity, and so we did not fit regression models to these data. Qualitative results were analysed by estimating the proportion of donors returning a reactive result in 180-day time bins, up to three years post-infection. When a donor had more than one observation in a single time bin, that donor contributed a fractional number to the numerator and one to the denominator. Proportions reactive and Wilson score 95% confidence intervals are reported.

We compared means and medians of antibody S/CO ratios in donors presumed to be on ART and not on ART at the time of donation to assess whether ART is associated with depressed S/CO, which would indicate a potential risk that a small proportion of donors with treated HIV may serorevert and, therefore, fail to be detected during blood screening. All analyses were conducted using the R statistical programming language (v 4.1.1, R Foundation for Statistical Computing, Vienna, Austria). 

## 3. Results

For the high-throughput blood screening assay testing of the MATHS early HIV treatment cohort, no clear pattern of declining signal intensity on any analyte was observed over time after ART initiation, in any of the treatment initiation groups (Figure 1). All three high-throughput blood-screening assays showed 100% detection in all time bins post-treatment initiation, in all three treatment initiation groups (Figure 2), indicating that no failure to seroconvert or seroreversions, were observed on these assays. 

The Abbott Determine POC assay also showed 100% detection rates in all time bins, with no seroreversion. However, the Abbott ABON HIV1 POC assay showed lower proportions detected in all time bins in the very early treated group, ranging from 77.1% (95% CI: 42.8–93.8%) to 83.1% (95% CI: 64.2–93.0%)—but with no dependence on time since treatment initiation. In the early and later treated groups, the assay showed high detection rates. The OraSure OraQuick assay showed poor detection rates in the early-treated group, ranging from 50.0% (95% CI: 26.8–73.2%) to 72.8% (95%CI: 53.2–86.3%), and, moderate detection rates in the early and later-treated groups. In the early treated group, the proportion detected showed some time dependence with, 92.7% detected in the first six months post-treatment initiation (95% CI: 74.5–98.2%), declining to 70.0% (95% CI: 29.9–92.7%) in the period from 900 to 1080 days post-treatment initiation.

For the HIV positive donations detected in 2017 and the HIV antibody+/NAT- donations from 2020, we observed higher serological signal intensities (signal to cut-off ratio, S/CO) in donors on ART compared to donors not on ART (Figure 3). On the Abbott PRISM, the median S/CO in treated donors was 101.1 (IQR: 84.8–117.8), and, in untreated donors was 85.1 (IQR: 57.2–103.7); on the Abbott Alinity, the median S/COs were 565.9 (IQR: 281.9–794.4), and 174.0 (IQR: 82.2–574.8) in treated and untreated donors, respectively. The medians are derived from the mean S/CO from triplicate testing of each donation sample.

## 4. Discussion

The absence of a pattern of declining reactivity on high-throughput screening assays, even in the very early treatment group, and 100% detection using these assays in all treatment groups, are reassuring. Our findings of inconsistent performance of rapid HIV tests are of concern, especially in resource-constrained settings where rapid tests may be used for blood donor screening or HIV diagnosis. 

In contrast to our findings, aborted seroconversion and seroreversion on high- throughput fourth-generation HIV antibody assays within months of treatment initiation have been documented in cohorts where treatment was initiated during acute infection (i.e., prior to seroconversion) [17]. In the MATHS cohort, treatment was not typically initiated within one to two weeks after infection, which may partly explain our different findings. Confirmation of laboratory results, contacting donors, and scheduling appointments resulted in a median delay of 18 days between index donation and enrolment. Blood samples taken at enrolment allowed reclassification of HIV disease stage at the time of study entry, by which time the number of early Fiebig stages had declined, resulting in 18 Fiebig stage I-III and 45 Fiebig stage IV-VI participants based on enrolment sample restaging. However, outside of acute treatment cohorts, such early initiation of ART is very uncommon, especially in South Africa where the vast majority of diagnoses are made using a rapid test algorithm, and acute infection is not frequently detected. Another reason for the difference between our findings and previous studies may be that we used fourth- and quasi-fifth-generation assays compared to previous studies that compared third- and fourth-generation assays [5]. These assays are not true quantitative assays but use the high dynamic range as a proxy for quantification. 

The use of POC rapid HIV tests is common practice for blood donation screening and HIV diagnosis in many Sub-Saharan African countries [18,19]. If rapid test screening is used, it is essential that a well-performing test be chosen, such as the Abbott Determine assay, which, in addition to demonstrating 100% detection rates in the very early treatment group, allows for p24 Ag detection, significantly reducing the pre-seroconversion window period in untreated individuals. However, it should also be noted that the Abbott Determine POC assay does not claim utility in screening of blood donors. 

The higher average S/CO from routine serological screening in donors on ART than in donors not on ART at the time of donation may reflect longer average durations of infection, given that very early treatment is likely still uncommon in this group, and, consequently, more mature antibody responses (increased avidity and/or affinity) would have developed by the time of HIV positive donation. This finding does not support the hypothesis that the increasingly early treatment initiation in South Africa—as treatment guidelines shifted to immediate ART initiation upon diagnosis—may have resulted in a greater proportion of donors exhibiting aborted seroconversion or seroreversion. This information does not raise blood safety concerns at present, although continued monitoring of this potential risk to the blood supply is essential. 

There are some limitations of the study. Firstly, we were unable to identify, enrol, treat, and test quickly enough in the MATHS cohort to ensure we had larger numbers in the Fiebig stage I and II group. As these were blood donors, the testing and confirmation was performed as part of the routine testing algorithms and the logistics to inform and counsel the RNA-positive, antibody-negative donors and then enrol and link them to care into a research study took longer than we expected. Therefore, it is quite possible that the assays would not have performed as well had the testing occurred sooner. Moreover, it is possible that a person who has a high-risk incident, starts treatment immediately and comes to donate blood could be missed. However, we believe this is highly unlikely. Secondly, the serological assays used in this study are not quantitative assays to estimate the antibody titre, but are qualitative assays in which we were interested in detection-sensitivity. Two of the high-throughput and one of the rapid POC assays detected antigen and antibody separately. In this study, we only analysed the antibody component of the assay, and it is possible that if antigen is detected the antibody detection is affected. A further limitation is that we did not evaluate the performance of rapid tests using the 2017 and 2020 cohort groups of donors. It could be possible that antiretroviral treatment initiation at any time may affect the POC assays’ sensitivity

The results of the current study as well as our previous discovery of persons with treated HIV presenting for donation have implications for transfusion safety [9]. The infectiousness of blood products derived from HIV infected but virally suppressed donors is not well-understood, but viral loads below the NAT assay’s limit of detection still result in substantial numbers of virions in the transfused product, and have been shown to be transmitted to the recipient [20]. Non-human primate studies have also confirmed that even small numbers of transfused virions can result in infection [21].

A remaining potential concern is undisclosed PrEP use, with rapidly growing uptake in high-risk groups, including adolescent girls and young women, who proactively protect their health. In South Africa, over half a million people are on PrEP [22]. Many of these potentially high-risk individuals would be deemed eligible to donate blood if risk behaviours were not disclosed or if the donors themselves had not engaged in any risky behaviour. Use of antiretroviral drugs for prevention of HIV by blood donors has been documented in the United States [10], and should suboptimal adherence or “on demand” PrEP use result in breakthrough infections followed by partial or complete viral suppression, the potential exists for infectious blood products to be released if both serological and NAT screenings fail. 

Blood collection organisations should consider the prevalence of HIV in the donor population, availability of PrEP, likelihood of very early ART initiation (especially ART initiation during acute infection), and the propensity of donors to disclose risky behaviour and prior HIV diagnoses and/or treatment. In addition, these organisations should also consider the costs and logistics when conducting risk assessments to determine whether a rapid test-based screening strategy sufficiently reduces the HIV transfusion and transmission risk.

While our findings are generally reassuring for blood safety in South Africa and other settings where major high-throughput serological screening assays are in use, caution should be exercised when relying on POC tests for donor screening and HIV diagnosis, and selection of well-performing POC tests is critical. Further study of emerging risks to the blood supply, including undisclosed ART and PrEP use, is required in order to monitor risk and ensure the continued safety of the blood supply.

## Figures and Tables

**Figure 1 viruses-14-01426-f001:**
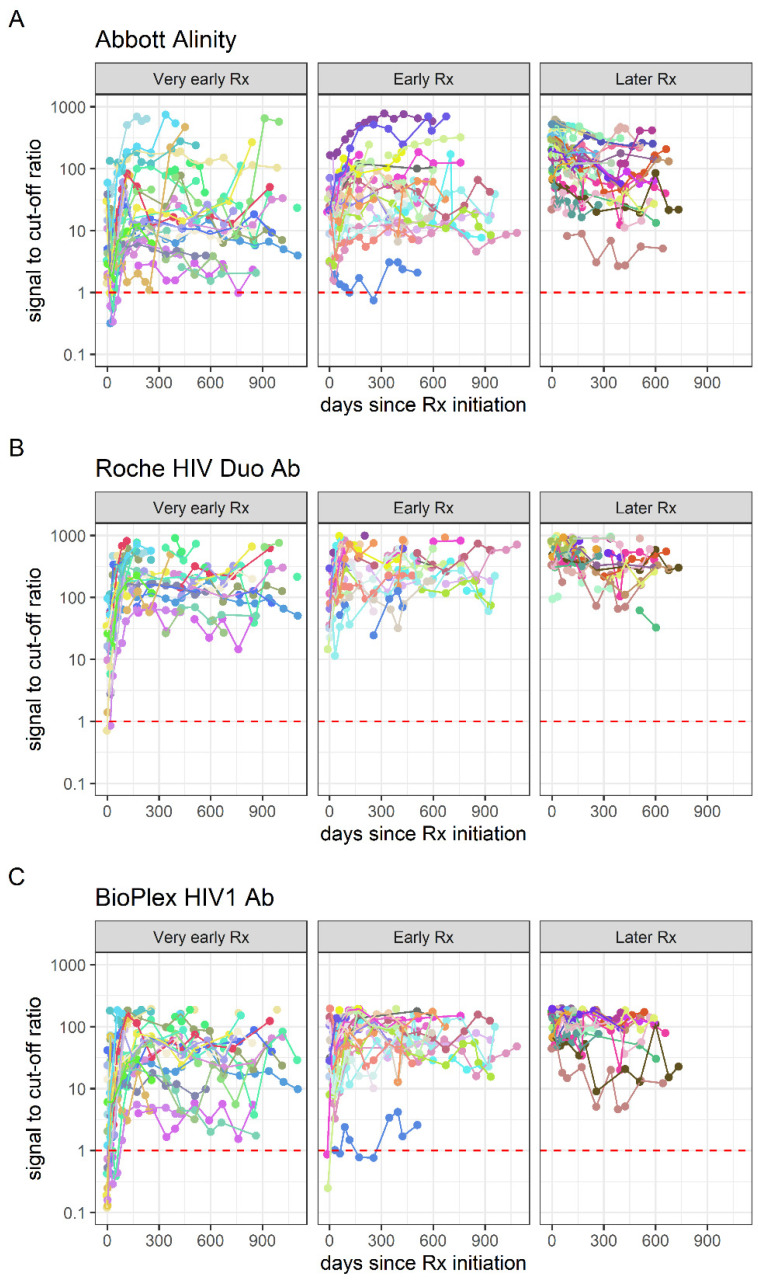
Quantitative signal intensity over time since treatment (Rx) initiation, stratified by time from infection to treatment initiation, in three high-throughput HIV blood screening assays. Panel (**A**): Abbott HIV Combo on the Alinity s platform, Panel (**B**): Roche Elecsys HIV Duo on the cobas e801 platform, and Panel (**C**): Bio-Rad BioPlex 2200 HIV assay. The dotted red lines indicate the cut- offs for positivity.

**Figure 2 viruses-14-01426-f002:**
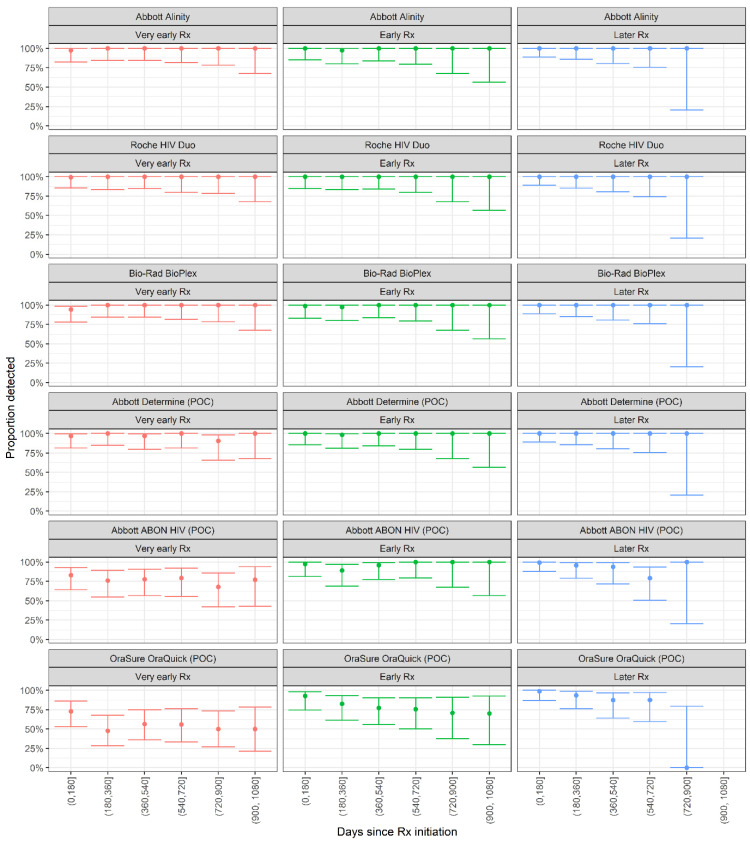
Proportion of donors returning reactive results on POC HIV antibody assays, stratified by 180-day intervals from infection to treatment (Rx) initiation. The points and whiskers indicate means and 95% confidence bounds.

**Figure 3 viruses-14-01426-f003:**
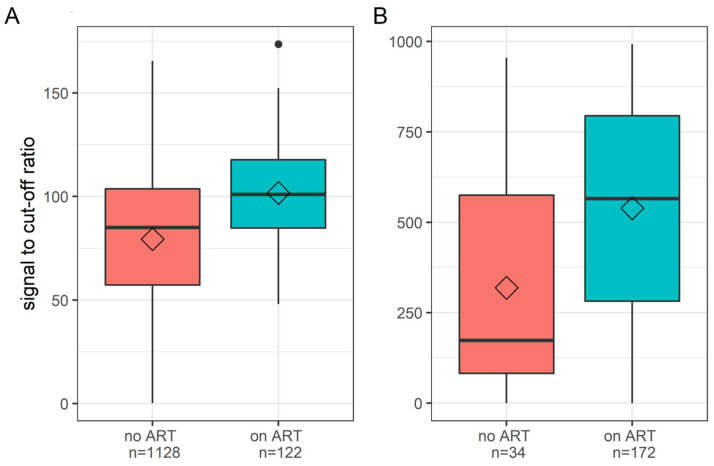
Distribution of signal to cut-off ratios in HIV positive samples in which ARVs were detected or not detected on the Abbott PRISM and Abbott Alinity s blood-screening systems. Panel (**A**): Abbott PRISM (third generation Ab assay); Panel (**B**): Abbott Alinity (4th-generation Ag/Ab combo assay). Horizontal lines in the boxes indicate medians, and, diamonds indicate means.

## Data Availability

The complete data set used in this study is stored at SACEMA.

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
