# Peer review of "The Impact of Early Antiretroviral Treatment (ART) for HIV on the Sensitivity of the Latest Generation of Blood Screening and Point of Care Assays"

_viruses, 2022, doi:10.3390/v14071426_

Round 1

Reviewer 1 Report

Comments Vermeulen et al:

The authors examined if there is any correlation between early ART initiation and seroconversion rate in HIV infected individuals of South Africa. The data did not show much correlation, as they found ubiquitous seroconversion. However, their main concern was due to early ART initiation following acute HIV infection; the system will not have sufficient exposure to generate adequate antibody response. Therefore, absence of HIV RNA due to ART and anti-HIV antibody due to early ART initiation will create problem in screaming blood. Especially, when persons with known HIV infection are allowed to donate blood while on ART, which raises potentially serious concerns for blood safety during transfusion. Given that, their data did not find any correlation between early ART initiation and seroconversion rate in HIV infected individuals of South Africa, the above concern does not exist as long as sensitive HIV detection tests are used for blood screening.

Major concern:

1) In its current form, text is not clear at certain places; thus, need English editing and elaboration for better/clearer explanation of results.

Please describe following in much detail for the better understanding of the non-expert reader:

  • 4th and 5th generation blood screening assays
  • Abbott ABON HIV1
  • point of care (POC) rapid tests
  • OraSure OraQuick POC assays

Author Response

Thank you for this recommendation. We have added the following two paragraphs under the heading of 2.5 Assays.

The Abbot Alinity is a 4th generation Ag/Ab combo assay that detects both antigen and antibody without the ability to discriminate between the two. The Roche HIV Duo and Biorad Bioplex 2200 are pseudo “5th generation” assays. The mechanism of detection is the same as the 4th generation assays however the antigen and antibody reactions are separate allowing for discrimination.

The rapid POC assays are not licensed for blood screening. The Abbott Determine detects antigen and antibody whereas the Abbott ABON HIV and Orasure Oraquick only detect antibody. These POC assays are immunochromatographic lateral flow assays that use a test strip that is pre-coated with HIV antigens.

Reviewer 2 Report

The manuscript by Vermeulen et al. reports on blood donors identified as early HIV-1 infections receiving ART monitored by antibody/combo assays exploring potential delay or absence of seroconversion. None of this seemed to occur.

In the introduction, the impact of Art on viral load should be described in addition to serology.

It is unclear how ‘acute and recent HIV infection’ is defined. Is it only pre-seroconversion window period?

I’m not sure 4th and 5th generation HIV tests is understood by readers. It should be made clear that they are combo antigen and antibody and what is the difference between 4th and 5th. It should be made also clear that POC tests are strictly detecting antibodies.

M&M

There should be a reference for the SBC LAg test and some explanation of why it indicates early infection. Monitoring IgM to HIV might be an alternative and more classical approach.

In this section, there is no mention of quantification of viral load. This marker would be critical for identification/classification of window period/early cases, the monitoring of ART and the detection of ART metabolites. Missing such VL quantification would be a serious weakness of the study.

Monitoring S/CO as indicating level of antibodies is very rough and poorly informative. Limiting dilution would be a more valuable approach. In addition, testing separately antibody levels of specific antibodies such as anti-core, anti-gp41 or anti-gp120 might be informative.

Results

It is this reviewer’s understanding that Abbott, Roche or Bio-Rad combo assays detect both antibodies and antigen. These assays might therefore be biased by antigen detection as monitoring antibody behaviour. In contrast, the three POCs detecting only antibodies might be a cleaner reflection of antibody levels.

In Figure 1, the test profile of each group (very early, early and late) is clearly different with increasing levels in the first, mostly stabilised in the second and declining in the third. Considering the uncertainties of classification, individuals with increasing test levels could be classified as early, examined together and compared with those stable and those declining. Instead of classifying according to time from ART, it might be productive to compare according to time from NAT positivity.

As to data presented in Figure 3, the criteria for ART initiation should be given. Is there a cut-off of viral load for ART initiation? Is the interval between diagnosis and study testing comparable? There might be other elements intervening to explain this difference observed with assays not truly quantitative.

Discussion

The initially intended objective of the study was whether or not early (in fact not early enough) initiation of ART in pre-seroconversion HIV-1 infection could prevent sero-conversion or induce sero-reversion. The answer is clearly no under the circumstances of the study that waited for 2 or more weeks before therapy and essentially defeated the purpose.

Later, the emphasis shifted to utility of different assays, a subject already largely examined and invariably concluding to lower sensitivity of POCs rapid tests although the impact on blood safety remains unclear.

The authors need to include in their discussion a section on study weaknesses that are many. As indicated above, viral load quantification over time is missing. The combos used can be or are biased by detection of viral antigen. There is no truly reliable quantification of antibodies. ART was delayed far too long to truly examine the initial study objective. In the end not much new or convincing data was generated.

Author Response

Thank you for taking the time to review the paper. i have responded in the attachment

Round 2

Reviewer 2 Report

The authors did not insert a paragraph listing and explaining the multiple weaknesses of this study listed in my review. Each of the listed issues should be clearly exposed and discussed. Ultimately, negative results have been obtained and reasons for such disappointing outcome need to be exposed.

Author Response

We gave added the following paragraph to the discussion

There are some limitations of the study. Firstly,  we were unable to test, identify, enroll, treat and test quickly enough in the MATHS cohort to ensure we had larger numbers in the Fiebig stage I and II group. As these were blood donors the testing and confirmation was performed as part of the routine testing algorithms and the logistics to inform and counsel the RNA positive, antibody negative donors and then enroll and link them to care into a research study took longer than we expected. It is therefore quite possible that the assays would not have performed as well had the testing occurred sooner. Therefore, it is possible that a person who has a high-risk incident, starts treatment immediately and comes to donate blood could be missed. However, we believe this is highly unlikely. Secondly, the serological assays used in this study are not quantitative assays to estimate the antibody titer, but are qualitative assays in which we were interested in detection sensitivity. Two of the high through put and one of the rapid POC assays detected antigen and antibody separately. In this study we only analyzed the antibody component of the assay, and it is possible that if antigen is detected the antibody detection is affected. A further limitation is that we did not evaluate the performance of rapid tests using the 2017 and 2020 cohort  groups of donors. It could be possible that antiretroviral treatment initiation at any time may affect the POC assays’ sensitivity